# Under the COVID-19 Environment, Will Tourism Decision Making, Environmental Risks, and Epidemic Prevention Attitudes Affect the People’s Firm Belief in Participating in Leisure Tourism Activities?

**DOI:** 10.3390/ijerph18147539

**Published:** 2021-07-16

**Authors:** Kuan-Chieh Tseng, Hsiao-Hsien Lin, Jan-Wei Lin, I-Shen Chen, Chin-Hsien Hsu

**Affiliations:** 1MA Program in Social Enterprise and Cultural Innovation Studies, College of Humanities & Social Sciences, Providence University, Taichung 43301, Taiwan; jackt72@pu.edu.tw; 2Department of Leisure Industry Management, National Chin-Yi University of Technology, Taichung 41170, Taiwan; chrishome12001@yahoo.com.tw (H.-H.L.); ishenc@hotmail.com (I.-S.C.); 3Center for General Education Sports Administrative Organization, Center for General Education, National Chi-Nan University, Puli 545, Taiwan; cwlin@ncnu.edu.tw

**Keywords:** COVID-19, Dajia Matsu pilgrimage, tourism decision making, epidemic prevention attitude

## Abstract

The study was conducted to understand the travel intentions of Dajia Matsu pilgrimage participants through tourism decision making, environmental risk perception, epidemic prevention attitude, and physical and mental health assessment. A questionnaire survey was used to collect 230 questionnaires in the field during the 2021 pilgrimage, and structural analysis was conducted using SPSS 26.0 and AMOS 20.0 statistical programs. The results showed that environmental risk and physical and mental health awareness were not significantly associated with the travel intention of Dajia Matsu pilgrimage participants (p > 0.05), while travel decision and attitude toward epidemic prevention were significantly associated with travel intention (p < 0.05).

## 1. Introduction

Tourism is a human behavior and phenomenon derived from attracting tourists to visit a place by virtue of its natural environment, ecological landscape, customs, and culture [1]. The tourism industry has always been one of the main sources of economic income in various countries [2], and it is also a way for people to obtain entertainment and relaxation [3]. Religious culture has a supernatural, unexplained attraction associated with the power of worship and faith [4]. Religious tourism is a type of tourism activity in which religious sites, monuments, or landscapes are used as resources for tourism appreciation, study, participation, and learning [5]. Typically, this type of tourism includes several types of religious pilgrimages, expeditions, sightseeing, and wellness tourism. For many years, religious and cultural tourism has been one of the main tourism features in various countries, attracting a large number of tourists to travel and spend money [6], which is one of the economic income sources for the development of the tourism industry.

Matsu is a traditional Chinese cultural custom and deity and is one of the beliefs of the Chinese people. Since its introduction to Taiwan in 1644, many people in Taiwan have come to believe in Matsu. Many temples, altars, houses, and boats have been built for this purpose [7]. To this day, the Matsu pilgrimage is organized every year in April in Taiwan. It is a religious pilgrimage that lasts for nine days and eight nights, spanning four counties and cities, with a round trip of about 340 km and a record number of visitors of over 2 million [8]. It is estimated to generate at least $1 billion USD in business opportunities [9], bringing huge economic benefits to the region.

However, since December 2019, the rise of the COVID-19 epidemic has spread globally, hitting the tourism industry in various countries [2,10]. Although vaccination has been initiated [11], there has been no respite yet [12]. Countries are trying to take precautions against the epidemic, expecting to resume tourism activities, revive the tourism market, and improve the plight of the tourism industry [13]. However, due to the uncertainty of the mode of infection and the high mortality rate of the epidemic [14], people’s willingness to travel remains low [15]. The market continues to be impacted by the failure to revive the tourism trend [16].

The successful experience of the Taiwanese government in managing the SARS outbreak gave the public confidence that the government could effectively combat the COVID-19 outbreak [17]. In addition, the government and religious event organizers in Taiwan proposed measures to prevent cluster infections from occurring during the event, causing a breach in epidemic prevention: live broadcast for non-participants to watch; no crawling under the sedan chair; no overnight stay in pilgrims’ buildings; no sharing of large pot meals; registering actual names when entering the temple; and wearing masks throughout the event [18]. They advocated that participants should maintain an epidemic-proof distance during meals, lodging, and interactions with strangers; carry a cell phone with them to keep track of epidemic prevention information; and avoid conversations during meals. They also advised participants to stay in legal lodgings, not to rub their eyes, not to touch their mouths and noses, and to avoid physical contact with strangers, to wash their hands regularly, to drink plenty of water, not to eat raw food, and to consume opened food as soon as possible [19]. The congregation also volunteered to prepare disinfection materials and epidemic masks for use by participants [20] and established epidemic control mechanisms with central and local health authorities in order to achieve the goal of zero breach of epidemic prevention at the pilgrimage [21,22]. The organizers also planned a live broadcast of the event for the public to watch voluntarily [23]. All of these actions affected the public’s trust in the government’s effectiveness in controlling the epidemic and impacted their confidence in participating in the campaign [24]. It is clear that some people were still skeptical about the decision-making process of epidemic prevention. Therefore, we believe that the level of decision-making cognition can be used to estimate the public’s approval of the epidemic prevention decision in the Dajia Matsu pilgrimage and further estimate the public’s intention.

Dajia Matsu is one of the beliefs of the people of Taiwan. Through pilgrimage activities, her followers pray for a smooth life and work as well as spiritual comfort and stability [3] in the hope of furthering their physical and mental health. However, despite seemingly sound epidemic prevention measures, there are still environmental risks associated with the Dajia Matsu pilgrimage. In particular, the COVID-19 epidemic has not yet been resolved, and there are unidentified routes of transmission in addition to droplets and contaminants from patients, resulting in high mortality rates [14,25]. Moreover, there have been frequent accidents, such as contusions caused by pushing and crowding, firecracker explosions, burns from burning gold paper, and traffic accidents [26,27,28]. As a result, there were certain human and external risks associated with the event, which caused uncertainty and affected people’s willingness to participate in the event [25]. Therefore, we believe that environmental risk perception can be used to estimate people’s willingness to travel.

Travel behavior is a subconscious plan that arises from the idea of the various physical or psychological needs of an individual [29]. The desire to travel occurs when people have a need for it. Although the government has been promoting autonomous epidemic prevention for many years and the organizers have been actively pushing for epidemic prevention measures, the willingness to travel is still undermined by the risk of infection and various accidents due to the epidemic [15]. Therefore, in order to motivate individuals to travel, in addition to strong beliefs, a high degree of acceptance of epidemic prevention decisions is required [30] to overcome psychological barriers. The attitude toward epidemic prevention refers to the degree of perceptual control over the implementation of a decision or a belief about a person, thing, or object [31]. It varies according to the level of perceived benefit to the individual’s physical and mental health [32]. Accordingly, believers can judge the risk of the upcoming pilgrimage activities as a basis for determining their willingness to participate in it [33]. Therefore, we believe that by predicting individual travel behavior through epidemic prevention attitudes [34], we can further analyze people’s attitudes toward epidemic prevention measures during the Dajia Matsu pilgrimage and help understand their travel intentions.

Furthermore, the main purpose of tourism activities is to provide participants with a phenomenon and behavior that improves their physical and mental health [29]. Nevertheless, under the threat of environmental infections, people’s participation in tourism behavior will constitute a risk [15]. For individuals, the risk of travel involves the risk of physical and mental health [35], which is incompatible with the purpose of travel. Failure to protect physical and mental health will affect the willingness to participate in travel [32] and further affect travel intentions and behaviors [36]. Therefore, we believe that conducting individual physical and mental health assessments can anticipate tourists’ willingness to participate in upcoming tourism activities and understand the public’s confidence in the epidemic prevention measures for the Dajia Matsu pilgrimage.

The severity of the COVID-19 epidemic [14], which has affected the overall economic and industrial development of the country [2,10] and threatened the quality of life and safety of the population [25], is the most important issue that needs to be addressed in all countries. When the situation is serious, faith is one of the ways to stabilize the people until there is a solution [4]. Taiwan is currently facing both epidemic threats and water scarcity [37]. The organizers held this event to engage the public in order to provide them with spiritual solace [4] and to boost the local economy so that the industry could recover [9]. Despite the various precautions planned by the government and organizers [18,19,20,21,22,23], uncertainties, such as the risk of epidemics and accidents [14], prevented the public from traveling [15]. The Dajia Matsu pilgrimage started as a faith-based event for the people of Taiwan [3] but over time has grown into a globally recognized mobile religious tourism event [8], with a historic record of approximately 4 million participants in 2021 [22]. Therefore, we believe that faith may attract participation despite travel and epidemiological risks and threats to one’s physical and mental health.

Therefore, we believe that it is necessary to develop scales to examine tourism decision making, environmental risk perceptions, epidemic prevention attitudes, physical and mental health, and travel intentions. In view of this, this study selected the variables related to tourism decision making, environmental risk perception, epidemic prevention attitude, physical and mental health assessment, and travel intention based on previous scholars’ works [15,25,30,31,32,35,36]. The validity and reliability of the scales were verified by confirmatory factor analysis, and the scales of tourism decision making and environmental risk perception, epidemic prevention attitude, physical and mental health, and travel intention were constructed. The results of the study are expected to provide a future reference for the related industries and researchers.

## 2. Literature Discussion

### 2.1. Tourism Decision-Making Cognition

The Dajia Matsu pilgrimage is an annual religious tourism event in Taiwan [21,22]. However, due to the lengthy duration and long journey, participants from different places need to be properly informed to plan their participation schedule and prepare for the expenses expected [18,21] in order to have the most comfortable and safe participation experience [23,26,27]. However, threats, such as the rapid spread of COVID19 and high lethality [32], have reduced people’s willingness to participate in outdoor activities [32] and created distrust in the safety of the outdoor environment [34]. Therefore, achieving safe and comfortable route planning in advance is an important issue so that the physical and mental health of the individual can be protected during participation.

The impressions obtained by personal perception and the viewpoint given to the environment or things through mental processes [37] are called cognition. Tourism policy cognition is the awareness of the tourism policy system and policy process and the perceptions after promotion [38]. This helps people to obtain safe and comfortable route planning [28] to protect their physical and mental health during participation [31,32]. In particular, with the current development of social networks, information is highly accurate and circulates quickly, which helps people to judge the truth of matters and improve their judgment in decision making [39]. People respond to matters in their surroundings by applying their innate behavioral patterns [40], and through the mental process of knowing and understanding things through consciousness activities, a feeling of the effectiveness of policy development is obtained through a process of perception, imagination, recognition, reasoning, and judgment [41].

From the literature, it is clear that tourism decision making is based on the development of tourism industries with local tourism resources to achieve the goal of ameliorating local and industrial development difficulties [38]. It is the local people who can best experience whether the desired improvements have been achieved and usually only after the policy has been implemented [42]. It can be discussed in terms of awareness of policy regulations and content, government planning and supporting measures, industry measures, personal policy recognition, and expected policy effectiveness [43]. The higher the level of policy recognition, the higher the chance of public participation [30,39].

### 2.2. Environmental Risk Perception

Environmental risk refers to the extent to which people or society as a whole are confronted with unpredictable but potentially far-reaching potential problems arising from environmental hazards [44], which contradict or even conflict with everyday knowledge [45]. The degree of an individual’s response to the risk is the environmental risk perception [46]. Large-scale religious activities were originally a source of support for the human heart. The Matsu pilgrimage, with its long duration and distant destinations, is a well-known international religious event in Taiwan [21,22], with participants from all over the world. However, the current threats, such as the rapid spread of COVID19 and the high mortality rate [10], have led people to worry about their health being affected by the environment and other people [43,47]. Therefore, the distrust of the environment creates a barrier to participation in activities [48,49].

It is evident from the literature that the uncontrollable factors of the epidemic have led to increased environmental risks of transportation, dining, and lodging for consumption [43,44,45,46] and that the safety of crowd interactions cannot be guaranteed [48], resulting in present or future commercial losses [49]. These are the main reasons for considering environmental risks when people participate in religious tourism activities [50].

Therefore, to understand people’s trust in the surrounding environment during the event, we can examine the transportation routes, rides, meals, accommodations, and interpersonal communication processes required to participate in the event [21,22,42,43,44,45,46,47,48,49], which can help to understand people’s perception of environmental risks during the event. Furthermore, this may have a negative impact when individuals have a high degree of uncertainty about future outcomes, and the higher the uncertainty about the future, the greater the risk [51,52,53].

### 2.3. Epidemic Prevention Attitude

Individuals’ reactions to objects and situations, such as relevant people, events, objects, or policies, through their life experiences and in a state of mental and neurological readiness are called attitudes [54]. The Matsu pilgrimage is a major religious and cultural event held annually in Taiwan with a large number of participants [21,22]. Because of the uncontrollable factors, such as the highly contagious and lethal nature of the epidemic [52], the public has become more aware of the importance of epidemic protection [14,15,16,17]. The attitude towards epidemic prevention mentioned in this study refers to the extent to which people agree with the decision to travel for epidemic prevention during their participation in the activity [53,54]. The process of absorbing others’ values to modify one’s existing values has a direct and dynamic impact [55].

It is evident from the literature that a high attitude towards epidemic prevention allows the public to appreciate achievements, value and recognize the practice, and look forward to having such necessary resources or complying with such measures [56]. Epidemic prevention attitudes can be a predictive approach [57] and can be characterized in terms of decisions that are effective in reducing risk and are wise, correct, important, and satisfactory in their effectiveness. Therefore, to understand the mindset of the public during the campaign, we can examine the understanding, trust, cooperation, and implementation of public prevention measures [17,34,42,43]. The higher the degree of acceptance, the higher the chance of participation [42,58,59].

### 2.4. Physical and Mental Health Assessment

A person’s physical and mental health is defined as a state of physical, psychological, and social well-being [60]. It is an assessment analysis of self-perceptions [61]. Scientific evidence such as self-assessment can be used to present the actual situation [62]. The higher the health risk, the greater the impact on the individual’s behavioral decisions [10,17]. However, this epidemic poses a great harm and threat to the physical and mental health of individuals [10,11,12,13,14,15,16], leading to a greater emphasis on their physical and mental health. Therefore, we believe that those who participate in the pilgrimage are not only seeking spiritual support but also value the protection of their physical and spiritual health.

From the literature, it is known that investigating the current state of physical and mental health of individuals by their feelings can present the impact of the current environment on people [10]. Physical and mental health can be divided into three dimensions: psychological, spiritual, and attitudinal [28,63,64], which are evidenced by feelings such as anxiety, competence, enthusiasm, headache, abdominal pain, insomnia, stomach pain, abnormal diet, and death-seeking ideation [65,66]. Feelings of physical and mental health affect the individual’s willingness to act and judgment [6,17]. Moreover, the better the individual’s perception of physical and mental health during the participation process, the higher the willingness to participate in the activity [60,61].

### 2.5. Travel Intention

Intentions are individuals’ tendencies to anticipate, plan, or intend whether or not to engage in future behavior [67] and can be a tool for predicting future behavior [68]. It can be considered as the extent to which people have a tendency to want to travel in order to fulfill their travel beliefs [69]. This can be used to determine the degree to which an individual is inclined to engage in travel behavior for a particular future travel activity [70].

The Matsu pilgrimage is a large scale, outdoor, migratory religious event with regular routes and destinations, just like the migration of animals [7,20,21]. Because of the lengthy duration, long distances, and large number of people, the event has generated various consumer demands and business opportunities, making it a kind of alternative leisure tourism activity [7]. By holding the event at a regular frequency, the public can plan or evaluate their participation in advance, thereby forming an evaluation mechanism for predicting personal participation in the event.

The literature shows that travel intention is mainly a measure to understand the degree to which individuals travel and consider traveling [71] and to explore the degree of behavioral propensity to travel to a place [72]. It can be explored in terms of the intention to travel, the level of information, the personal behavior, and the necessary expertise required to prepare for the travel behavior in advance [72,73].

## 3. Method

### 3.1. Research Structure

Matsu is one of the main beliefs of the people of Taiwan. Believers look forward to spiritual solace and pray for their career and health by participating in the pilgrimage [3]. However, the outbreak has not yet improved [14,23]. Although epidemic prevention decisions may seem sound, there are still risks associated with participation in large gatherings [26,27,28], which in turn affects travel intentions [23]. Therefore, this study suggests that an examination of travel decisions, environmental risk perceptions, epidemic prevention attitudes, and physical and mental health perceptions [17,28,29,30,31,32,33,34,35,36,37,38,39,40,41,42,43,44,45,46,47,48,49,50,51,52,53,54,55,56,57,58,59,60,61,62,63,64,65,66] can help estimate people’s willingness to travel [67,68,69,70,71,72,73]. Therefore, this study will deduce the willingness to participate in tourism in terms of tourism decision, environmental risk perception, attitude toward epidemic prevention, and physical and mental health perception through the study of the Dajia Matsu pilgrimage in order to provide recommendations for the decision making of the government and related organizations that hold large-scale cluster events during the COVID-19 epidemic. The structure of the study is shown in Figure 1.

### 3.2. Research Hypothesis

According to the description of the research framework, four research hypotheses are proposed:

**Hypothesis** **1** **(H1).**
*Tourism decision-making cognition and tourism willingness have a significant impact.*


**Hypothesis** **2** **(H2).**
*Assumes that environmental risk perception and travel willingness have a significant impact.*


**Hypothesis** **3** **(H3).**
*Assumes that preventive attitudes and willingness to travel will have a significant impact.*


**Hypothesis** **4** **(H4).**
*Assumes that physical and mental health and willingness to travel will have a significant impact.*


### 3.3. Research Object

We first collected information related to the Dajia Matsu pilgrimage and epidemic prevention policies [1,2,3,4,5,6,7,8,9,10,11,12,13,14,15,16,17,18,19,20,21,22] and reviewed the literature on the effects of travel decision making, environmental risk perception, epidemic prevention attitude, and physical and mental health assessment on travel intentions [28,29,30,31,32,33,34,35,36,37,38,39,40,41,42,43,44,45,46,47,48,49,50,51,52,53,54,55,56,57,58,59,60,61,62,63,64,65,66,67,68,69,70,71,72,73]. A questionnaire survey was conducted with the participants of the 2021 Dajia Matsu pilgrimage. The travel intentions of the participants of the 2021 Dajia Matsu pilgrimage were examined using a research framework that included the components of travel decision making, environmental risk awareness, epidemic prevention attitudes, physical and mental health, and travel intentions. The authors participated in the 2021 Dajia Matsu pilgrimage from April 29 to April 18 and distributed 300 questionnaires at the event. A total of 230 valid questionnaires were collected, with 76.6% valid questionnaires. Lastly, data from the valid questionnaires were coded and entered into IBM SPSS Statistics 26.0 statistical software (IBM Corp, Armonk, NY, USA.) for narrative analysis of demographic variables, followed by the analysis of the relationship between variables and validation of the study model using AMOS 20.0 software (IBM Corp, Chicago, IL, USA).

### 3.4. Research Tools

The study aimed to investigate the factors influencing the travel intentions of the participants in the Dajia Matsu pilgrimage in terms of the relationship between travel decision-making cognition, environmental risk perception, epidemic prevention attitude, and physical and mental health assessment on travel intentions. A total of 39 questions were developed with reference to studies on travel decision-making cognition [17,38], environmental risk perception [49,50], epidemic prevention attitude [56,57], physical and mental health assessment [17,63,64], and travel intention [69,70,71,72].

Demographic variables included gender (male, female), age (under 20, 21–30, 31–40, 41–50, 51–60, 61+), and marriage (single/other, married). The questions were answered on a five-point Likert scale and were categorized into “strongly disagree,” “disagree,” “average,” “agree,” and “strongly agree” according to the intensity of feelings and were given scores of 1, 2, 3, 4, and 5, respectively. The questions are described in Table 1 below.

### 3.5. Data Processing and Analysis

After recovering the questionnaires and eliminating the invalid ones, the data were entered into SPSS 26.0 statistical software for narrative analysis of demographic variables, and then AMOS 20.0 software was used to analyze the relationship between equal variables and to verify the rationality of the study model, as shown in Figure 2, Figure 3, Figure 4, Figure 5 and Figure 6.

## 4. Research Result

### 4.1. Sample Descriptive Statistics 

The results of the analysis based on 230 valid questionnaires showed that 50.9% of the participants were male and 49.1% were female. In terms of age, 10.4% were under 20 years old, 16.5% were 21–30 years old, 16.1% were 31–40 years old, 14.8% were 41–50 years old, 25.2 were 51–60 years old, and 17% were over 61 years old. Single participants accounted for 44.8%, and married participants accounted for 55.2%, as shown in Table 2.

It can be seen that the Dajia Matsu pilgrimage is very attractive to the public, resulting in little gender difference in participants. Although 42.2% of the participants were over 51 years old, there was little difference in participation by age. Married people were more enthusiastic about participating in the Dajia Matsu pilgrimage.

### 4.2. Offending Estimates

According to Hair, Anderson, Tatham, and Black (1998), there are three cases in which offending estimates occur: (1) The error variance is negative. (2) The standardized regression coefficient is greater than 0.95. (3) The variance of the measurement error is not significant [74].

When performing model fitting, it is important to check for offending estimates. Offending estimates refer to estimated parameters that are outside the acceptable range for statistical analyses in structural equation models or measurement models. If offending estimates are present in the variables, the measurement problem must be addressed first before reviewing the parameter estimates.

As shown in Table 3, the absolute values of the standardized regression coefficients for people’s travel decision-making cognition, environmental risk perception, epidemic prevention attitudes, physical and mental health assessment, and travel intention ranged from 0.17 to 0.92 and did not exceed 0.95. The values of the error variance ranged from 0.02 to 0.12, with no negative error variance and no significant offending estimates; therefore, the fitness of the model can be examined.

### 4.3. Reliability and Validity Analysis

Understanding the main factors of participants’ willingness to participate in the 2021 Dajia Matsu pilgrimage is the main direction of this study. We attempted to obtain answers based on literature related to tourism decision-making cognition [28,31,32,39,40,41,42,43], environmental risk perception [21,22,42,43,44,45,46,47,48,49,50], epidemic prevention attitude [17,34,42,43], physical and mental health assessment [28,63,64], and travel intention [71,72,73].

In order to pursue a more accurate direction of inquiry, the present study utilized confirmatory factor analysis to measure the convergent validity and the construct validity of the questionnaire. Confirmatory factor analysis was used to gauge the convergent validity and construct validity of the questionnaire as part of SEM analysis. Too large CFA chi-square values can be corrected by using AMOS correction indicators [75]. A high MI value (greater than 3.84) indicates a measurement correlation between questions, and therefore, questions with high MI values were removed [76]. In this study, the MI values of 19 questions, including questions B3, B4, B6, B7, C1, C2, C7, C8, D4, D5, EE1, EE2, EE3, EE4, EE5, EE7, EE10, EE13, and F4, were too high, so they were deleted. The rest of the questions did not have excessive MI values and were therefore retained.

In the measurement model analysis, there are two issues that need to be examined: first, whether each observed variable is correctly measured for each potential variable in the overall model; and second, whether the construct validity in the validated model is convergent by examining whether the observed variables of different potential variables are loaded or not [76]. Generally, a standardized regression coefficient of greater than 0.7 is required for the measured variables to be considered as having convergent validity.

The average variance extracted for all measures of the potential variables was calculated as the average variance extracted for the observed variables, which reflects the average explanatory power of the potential variables. In this study, the mean extraction of variance was above 0.5, which is the standard [74], so this study has convergent validity.

It is generally believed that when measuring the component reliability of a potential variable, the greater the component reliability, the better the internal consistency of the measure and the more the construct validity of the potential variable can be demonstrated [77]. The results of this study showed that the component reliability values of all the components were above 0.6, which is in accordance with the requirements [77,78], so the internal quality of this model is good.

The Cronbach’s alpha test for each construct in this study also met the criteria [79,80], so the scales in this study have good reliability.

### 4.4. Convergent Validity

The convergent validity of the confirmatory factor analysis showed that the three indicators of factor loadings, mean variance extracted, and component reliability met the criteria [75,80,81,82], as shown in Table 4.

### 4.5. Overall Model Analysis

The structural model analysis is generally modified as suggested by Bollen [75], and the overall model fitness is examined using seven indicators, such as the χ^2^ test, the ratio of χ^2^ to degrees of freedom, GFI, AGFI, RMSEA, CFI, and PCFI. As can be seen from Table 3, the corrected χ^2^ to degrees of freedom ratio was 2.33 (<3), the value of GFI was 0.90 (>0.90), the value of AGFI was 0.82 (>0.80), the value of RMSEA was 0.07 (<0.08), the value of CFI was 0.94 (>0.90), and the value of PCFI was 0.77 (>0.50).

Furthermore, TLI (Tucker–Lewis index) is an index to determine the goodness of fit, which ideally should be between 0.8 and 0.95; SRMR (standardized root mean square residual) is the mean of the difference between the predicted matrix and the sample matrix, and its value should be lower than 0.08, and the smaller the value, the better. The analysis showed that the TLI and SRMR of the study framework were 0.93 and 0.0605, respectively, which are both within the reasonable range, and therefore, the data from this analysis were used for the study. The results of the analysis indicated that this model is acceptable, as shown in Table 5.

### 4.6. Validation of Research Hypotheses

As shown in Table 6, the four hypotheses listed in Figure 1 concerning tourism decision-making cognition, environmental risk perception, epidemic prevention attitude, physical and mental health assessment, and travel intention and their respective path values are: tourism decision-making cognition has a significant effect on travel intention, with a path value of 0.37; environmental risk perception does not have a significant effect on travel intention, with a path value of −0.01; epidemic prevention attitude has a significant effect on travel intention, with a path value of 0.20; and physical and mental health assessment does not have a significant effect on travel intention, with a path value of 0.08. As shown in Figure 7.

This analysis suggests that research hypothesis 1 is valid, that is, the public’s perceptions of tourism decisions have a significant effect on willingness to participate in tourism, a result that is consistent with previous studies [17,37]. The epidemic has not yet been effectively resolved, and there is uncertainty about the mode of infection and the environmental threat of high mortality [14]. However, the Dajia Matsu pilgrimage is a well-known religious and tourism event in Taiwan, and the government and organizers have well-planned epidemic prevention measures in place [17,18,19,20,21,22,23]. Although the public may still have concerns, the thoroughness of the epidemic prevention policies affects the willingness to participate in the parade. As a result, there is a significant positive correlation between people’s tourism decision-making cognition and travel intention.

Hypothesis 2 of the above analysis is not valid, i.e., people’s awareness of environmental risks in tourism does not have a significant effect on their willingness to participate in tourism, which is different from the results of previous studies [50,51,52,53]. It is inferred that although tourism risks and epidemics in tourism activities affect people’s willingness to participate, the Dajia Matsu pilgrimage is an annual religious event in Taiwan, and the government, companies, organizers, cultural organizations, and the public maintain a high level of recognition of the event and have a tacit agreement to control the epidemic [21]. In addition, the government and organizers have advocated for epidemic prevention, and the attitude of individuals towards epidemic prevention has been improved [20]. As a result, there is no correlation between people’s environmental risk perception and their travel intentions.

Hypothesis 3 of the above analysis is valid, that is, people’s epidemic prevention attitude has a significant effect on their willingness to participate in travel, which is the same as the results of previous studies [56,83]. It is inferred that the current epidemic problem seriously affects people’s willingness to go out for leisure activities [15], but travel can help people relax physically and mentally. The Dajia Matsu pilgrimage is one of the major faith-based events in Taiwan, and the epidemic control in Taiwan has been stable, and the organizers have taken proper measures to prevent epidemics [18,19,20,21,22]. Therefore, people believe that the effectiveness of epidemic control will be the key to whether or not to participate in the event. As a result, there is a significant positive correlation between people’s epidemic prevention attitude and their travel intentions.

Hypothesis 4 of the above analysis is not valid, i.e., people’s physical and mental health assessment does not have a significant effect on their willingness to participate in tourism, which is different from the results of previous studies [10,17]. Tourism activities have a stress-relieving effect on participants, and the Dajia Matsu pilgrimage is one of the major faith-based activities in Taiwan. Especially in the midst of a serious epidemic, people seek spiritual comfort from this event and pray for the disaster to be contained and safe and stable life to be restored. Therefore, people do not believe that their perceptions of their current physical and mental health had an impact on their willingness to participate in the pilgrimage. As a result, there was no correlation between people’s physical and mental health assessment and their travel intentions.

## 5. Conclusions

The Dajia Matsu pilgrimage is an annual religious event in Taiwan that is highly recognized by the government, corporations, organizers, civil society organizations, and the public. During the procession, people can find solace in their hearts and souls but may face uncertain travel risks. Sound epidemic prevention measures and travel decisions as well as people’s personal attitudes toward epidemic control will be the main influencing factors on travel intentions for participation in the Dajia Matsu pilgrimage.

### 5.1. For Research Objects

Taiwan has a wide variety of religious activities and outdoor events. Tourism events require the cooperation and input of decision makers (organizers), participants, and local residents in order to be successfully completed. Since this paper is based on the participants of the Dajia Matsu pilgrimage, it is not possible to compare the views of the participants of other events. Therefore, it is recommended that surveys be conducted on the perceptions of participants in other events to obtain more comprehensive answers.

### 5.2. For the Research Scope

Because of the diverse themes of religious activities in Taiwan and the abundance of large outdoor events, the differences in the attractiveness of each region or theme and the differences in timing may have different effects on the willingness to participate. Therefore, we believe that it is not possible to compare the perceptions of participants in other activities by using the Dajia Matsu pilgrimage as the theme of this study. Therefore, it is recommended that surveys be conducted on the participants of other outdoor activities or Pilgrimage events as well as on the differences in the timing of these events in order to obtain more complete answers.

### 5.3. For Detour Activities

According to the results of this study, tourism decision making and epidemic prevention attitudes were the main factors influencing the travel intentions of the participants in the event. Therefore, in the future, when planning the event, the organizers should make sound tourism decisions, enhance participants’ sense of recognition, and promote participants’ epidemic prevention attitudes so that more people will recognize and participate in the event.

## Figures and Tables

**Figure 1 ijerph-18-07539-f001:**
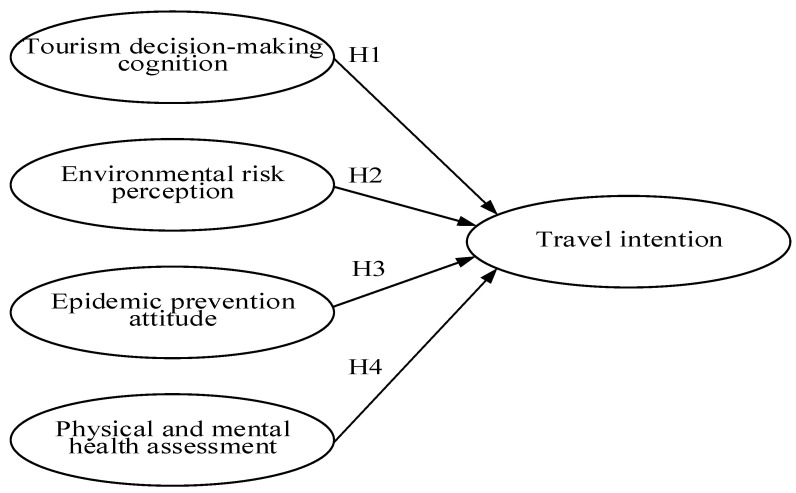
The research structure.

**Figure 2 ijerph-18-07539-f002:**
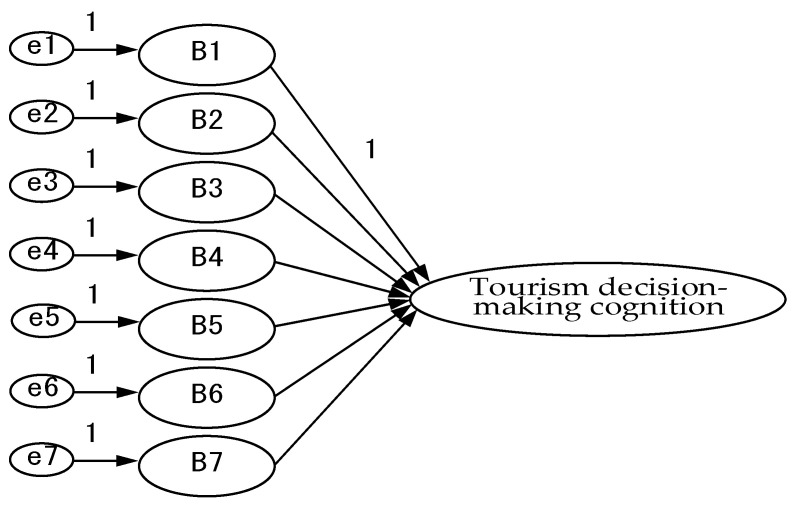
Tourism decision-making cognition model—confirmatory factor analysis framework.

**Figure 3 ijerph-18-07539-f003:**
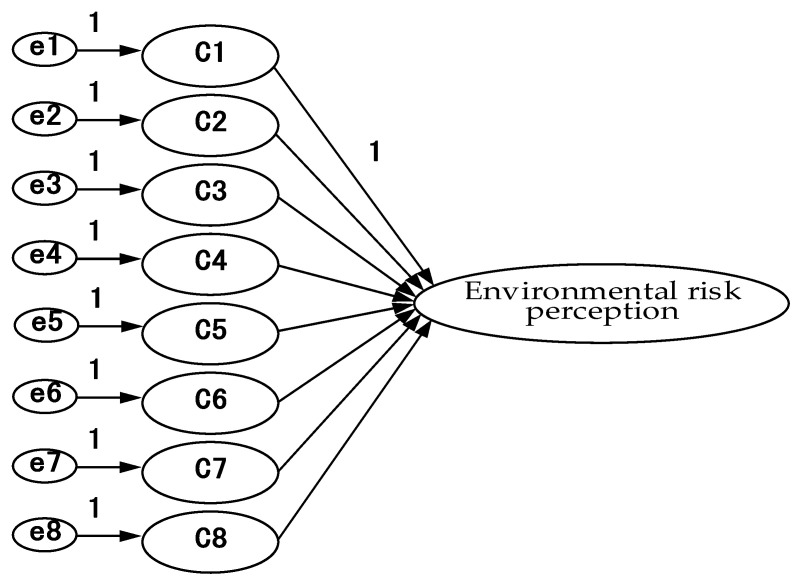
Environmental risk perception model—confirmatory factor analysis framework.

**Figure 4 ijerph-18-07539-f004:**
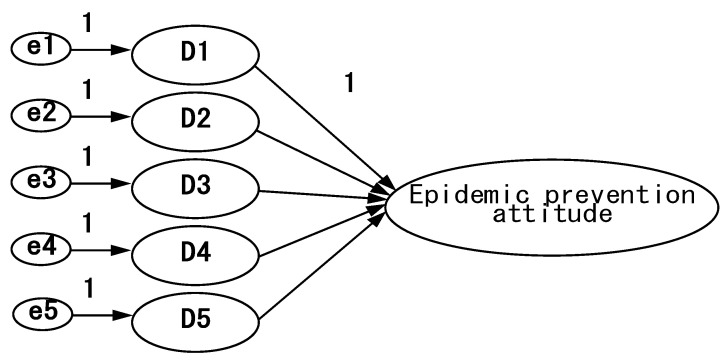
Epidemic prevention attitude model—confirmatory factor analysis framework.

**Figure 5 ijerph-18-07539-f005:**
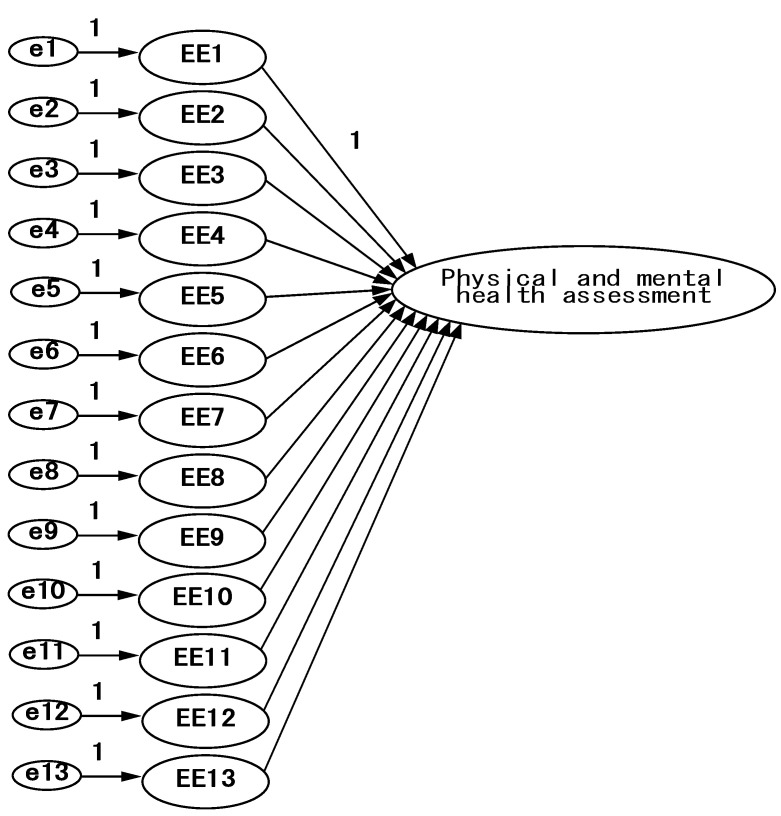
Physical and mental health assessment model—confirmatory factor analysis framework.

**Figure 6 ijerph-18-07539-f006:**
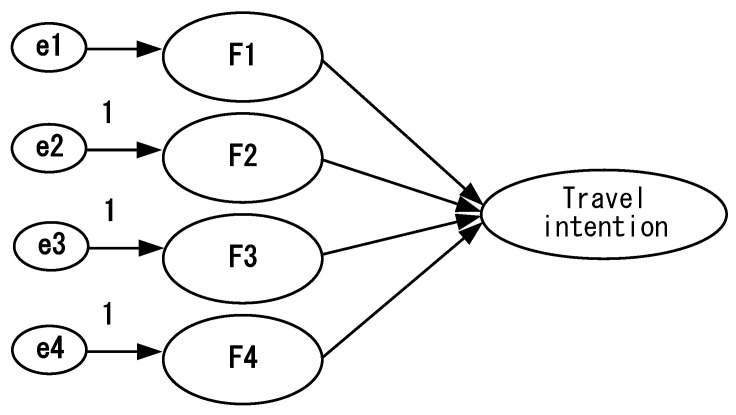
Travel intention model—confirmatory factor analysis framework.

**Figure 7 ijerph-18-07539-f007:**
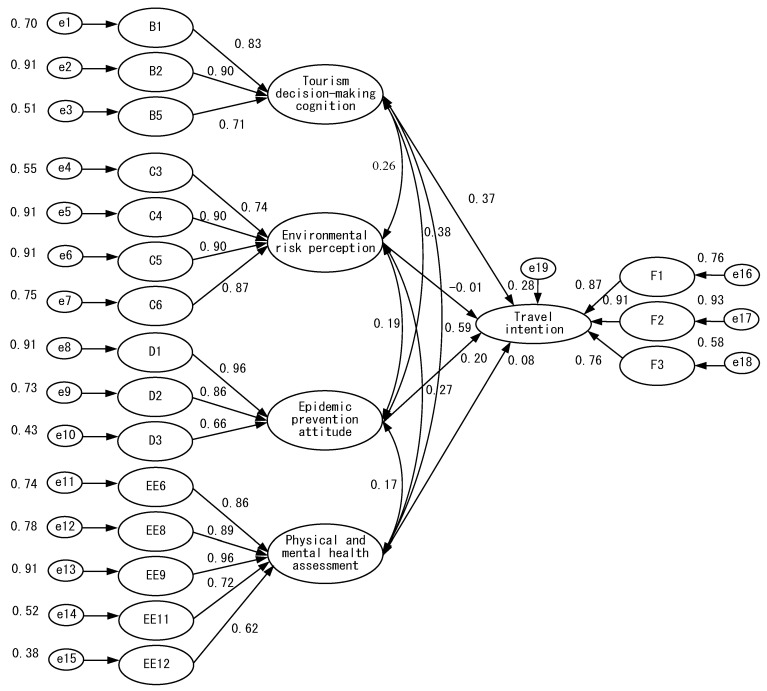
The model of travel intention.

**Table 1 ijerph-18-07539-t001:** Tourism decision-making cognition, environmental risk perception, epidemic prevention attitude, and physical and mental health assessment on travel intention.

	Issue
Basic Variables	Gender (male, female); age (20 down, 21–30, 31–40, 41–50, 51–60, Over 61); marriage (single/other, married)
Tourism decision-making cognition (B)	B1. Agree with tourism policies; B2. Agree with relevant epidemic prevention and tourism planning; B3. You still have to do your own epidemic prevention measures; B4. Related subsidy measures; B5. Detour is a proper tourism planning decision after evaluation; B6. Can create domestic tourism benefits; B7. It can improve the plight of the domestic tourism industry.
Environmental risk perception (C)	C1. Airport transfer may be infected with COVID-19; C2. COVID-19 may be transmitted when taking public transportation; C3. COVID-19 may be transmitted when entering and leaving indoor tourist places; C4. COVID-19 may be transmitted when entering and leaving outdoor tourist places; C5. You may be infected with COVID-19 while eating or while in your accommodations; C6. Shopping and consumption may transmit COVID-19; C7. People around you avoid participating in detour activities; C8. COVID-19 is a frightening disease.
Epidemic prevention attitude (D)	D1. It is wise to take preventive measures in advance; D2. It is correct to take preventive measures in advance; D3. It is important to me to take preventive measures in advance; D4. Government preventive measures can effectively reduce risks; D5. I am satisfied with government preventive measures.
Physical and mental health assessment (EE)	E1. Feel good and increase confidence; E2. Reduce feelings of fear; E3. Increase job performance satisfaction; E4. Be enthusiastic about things or activities; E5. Increase work efficiency; E6. Relieve headaches or overhead pressure; E7. Reduce backache problems; E8. No more insomnia or a good night’s sleep; E9. No stomachache and indigestion; E10. Recover appetite; E11. No more anxiety and tantrums; E12. I feel that work and life are meaningless; I feel very lost; E13. Death caused by the depression of life or the idea of escaping from everything.
Travel intention (F)	F1. Willing to participate in free time; F2. High willingness to participate in the future; F3. Continue to collect detour information; F4. Try to improve individual participation conditions.

**Table 2 ijerph-18-07539-t002:** Analysis of demographic variables.

Background	N	%
Male	117	50.9%
Female	113	49.1%
Under 20	24	10.4%
21–30	38	16.5%
31–40	37	16.1%
41–50	34	14.8%
51–60	58	25.2%
Over 61	39	17.0%
Single//other	103	44.8%
Married	127	55.2%

**Table 3 ijerph-18-07539-t003:** Offending estimates of the scales for tourism decision-making cognition, travel environmental risk perception, epidemic prevention attitude, physical and mental health assessment, and travel intention.

Item	Standardized Regression Coefficient	Deviation Variance
B1<---Tourism decision-making cognition	0.82	0.05
B2<---Tourism decision-making cognition	0.84	0.04
B3<---Tourism decision-making cognition	0.58	0.04
B4<---Tourism decision-making cognition	0.64	0.08
B5<---Tourism decision-making cognition	0.79	0.04
B6<---Tourism decision-making cognition	0.64	0.04
B7<---Tourism decision-making cognition	0.25	0.07
C1<---Environmental risk perception	0.35	0.08
C2<---Environmental risk perception	0.68	0.04
C3<---Environmental risk perception	0.79	0.02
C4<---Environmental risk perception	0.87	0.03
C5<---Environmental risk perception	0.89	0.02
C6<---Environmental risk perception	0.87	0.03
C7<---Environmental risk perception	0.41	0.12
C8<---Environmental risk perception	0.17	0.08
D1<---Epidemic prevention attitude	0.92	0.02
D2<---Epidemic prevention attitude	0.88	0.02
D3<---Epidemic prevention attitude	0.66	0.03
D4<---Epidemic prevention attitude	0.53	0.06
D5<---Epidemic prevention attitude	0.39	0.09
EE1<---Physical and mental health assessment	0.73	0.05
EE2<---Physical and mental health assessment	0.79	0.04
EE3<---Physical and mental health assessment	0.76	0.04
EE4<---Physical and mental health assessment	0.69	0.04
EE5<---Physical and mental health assessment	0.73	0.05
EE6<---Physical and mental health assessment	0.90	0.03
EE7<---Physical and mental health assessment	0.35	0.46
EE8<---Physical and mental health assessment	0.86	0.04
EE9<---Physical and mental health assessment	0.91	0.03
EE10<---Physical and mental health assessment	0.87	0.03
EE11<---Physical and mental health assessment	0.72	0.06
EE12<---Physical and mental health assessment	0.62	0.08
EE13<---Physical and mental health assessment	0.62	0.09
F1<---Travel intention	0.83	0.03
F2<---Travel intention	0.91	0.03
F3<---Travel intention	0.81	0.06
F4<---Travel intention	0.77	0.06

**Table 4 ijerph-18-07539-t004:** Convergent validity and construct reliability of this study.

Issue	Standardized Loadings	Unstandardized Loadings	S.E.	C.R. (t-value)	p	SMC	C.R.	AVE
B1<---Tourism decision-making cognition	0.84	1.00	-	-	-	0.71	0.85	0.66
B2<---Tourism decision-making cognition	0.89	0.95	0.07	13.27	***	0.80	-	-
B5<---Tourism decision-making cognition	0.71	0.69	0.06	11.46	***	0.50	-	-
C3<---Environmental risk perception	0.74	1.00	-	-	-	0.54	0.91	0.73
C4<---Environmental risk perception	0.90	1.58	0.12	13.62	***	0.81	-	-
C5<---Environmental risk perception	0.90	1.46	0.10	13.94	***	0.81	-	-
C6<---Environmental risk perception	0.87	1.54	0.12	13.26	***	0.76	-	-
D1<---Epidemic prevention attitude	0.98	1.00	-	-	-	0.96	0.87	0.69
D2<---Epidemic prevention attitude	0.84	0.87	0.06	14.25	***	0.70	-	-
D3<---Epidemic prevention attitude	0.65	0.69	0.06	10.59	***	0.42	-	-
EE6<---Physical and mental health assessment	0.85	1.00	-	-	-	0.73	0.90	0.66
EE8<---Physical and mental health assessment	0.88	1.10	0.06	17.89	***	0.78	-	-
EE9<---Physical and mental health assessment	0.96	1.10	0.05	20.92	***	0.93	-	-
EE11<---Physical and mental health assessment	0.72	0.87	0.07	12.87	***	0.52	-	-
EE12<---Physical and mental health assessment	0.62	0.75	0.07	10.53	***	0.39	-	-
F1<---Travel intention	0.87	1.00	-	-	-	0.75	0.88	0.72
F2<---Travel intention	0.92	1.10	0.07	15.92	***	0.84	-	-
F3<---Travel intention	0.76	1.15	0.09	13.51	***	0.58	-	-

*** *p* < 0.001.

**Table 5 ijerph-18-07539-t005:** Overall fitness analysis of the study model.

Fitness Indicator	Acceptable Criteria	After Mode Revision	Model Fitness Judgment
χ^2^ (Chi-square)	The smaller the better	291.33	-
Ratio of χ^2^ to degrees of freedom	<3	2.33	Fit
GFI	>0.90	0.90	Fit
AGFI	>0.80	0.82	Fit
RMSEA	<0.08	0.07	Fit
CFI	>0.80	0.94	Fit
PCFI	>0.50	0.77	Fit
TLI	0.80–0.95	0.93	Fit
SRMR	>0.80	0.0605	Fit

**Table 6 ijerph-18-07539-t006:** Study hypotheses and validation results.

Research Hypothesis	Path Value	Validation Results
Hypothesis 1: People’s perception of tourism decision-making has a significant impact on their willingness to participate in tourism.	0.37 *	Valid
Hypothesis 2: People’s perception of tourism environmental risks does not have a significant impact on their willingness to participate in tourism.	−0.01	Invalid
Hypothesis 3: People’s attitudes towards epidemic prevention have a significant impact on their willingness to participate in tourism.	0.20 *	Valid
Hypothesis 4: People’s physical and mental health assessment does not have a significant impact on the willingness to participate in tourism.	0.08	Invalid

* *p* < 0.05.

## Data Availability

Considering the privacy rights of interviewees, research data are not disclosed.

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
