# Peer review of "Under the COVID-19 Environment, Will Tourism Decision Making, Environmental Risks, and Epidemic Prevention Attitudes Affect the People’s Firm Belief in Participating in Leisure Tourism Activities?"

_ijerph, 2021, doi:10.3390/ijerph18147539_

Round 1
Reviewer 1 Report
The topic is interesting. However, there are too many hypotheses needed to be investigated in one report later. The pandemics are related to several potential variables and maybe the authors need to review them before comparing the proposed factors.
Second, for the types of the Journal (https://www.mdpi.com/journal/ijerph/instructions), “Case reports present detailed information on the symptoms, signs, diagnosis, treatment (including all types of interventions), and outcomes of an individual patient. Case reports usually describe new or uncommon conditions that serve to enhance medical care or highlight diagnostic approaches.” It seems that this article does not fit the form of the case report very well.
Taichung postcode?
“[20]. The organizers also planned a live broadcast of the event for the public to watch it voluntarily [22]. ” Citation [21]?
Figure 1-5: Please check the layout of the figures.
Author Response
Review 1
1.The topic is interesting. However, there are too many hypotheses needed to be investigated in one report later. The pandemics are related to several potential variables and maybe the authors need to review them before comparing the proposed factors.
Dear reviewer: Thanks for the reminder, we found an error and have corrected it.
2.Second, for the types of the Journal (https://www.mdpi.com/journal/ijerph/instructions), “Case reports present detailed information on the symptoms, signs, diagnosis, treatment (including all types of interventions), and outcomes of an individual patient. Case reports usually describe new or uncommon conditions that serve to enhance medical care or highlight diagnostic approaches.” It seems that this article does not fit the form of the case report very well.
Dear reviewer: It is set as the case report type because the manuscript is aimed at a special case, that is, a phenomenon, so this research was initially designated as the case report description.
However, thanks for the reminder, we agree with you, the manuscript tends to be Articles, and if necessary, we agree to amend it to Articles.
3.Taichung postcode?
Dear reviewer: Thanks for the reminder, the information has been added.
4.“[20]. The organizers also planned a live broadcast of the event for the public to watch it voluntarily [22]. ” Citation [21]? “[20]。
Dear reviewer: Thanks for the reminder, the order of the reference numbers has been rechecked.
5.Figure 1-5: Please check the layout of the figures.
Dear reviewer: Thanks for the reminder, the picture has been rearranged.
Reviewer 2 Report
The topic of the study is interesting, however, the paper left me a number of questions throughout the reading. Overall, the paper needs to improve overall coherence and rationales of the research.
First, I don't see the match of the title with contents and the research purpose.
Second, the rationals of the research is very weak. I don't see the theoretical background and literature review is appropriate and clear in order to support the research model. For example, the authors brought tourism decision-making cognition of tourism policy decision. The study focused on travel intention of participation of the 2021 Dajia Matsu pilgrimage. Then it is better to start with theories of travel decision-making process of travelers. The authors fail to address why tourism decision-making cognition of policy is needed rather than traveler's decision making theory.
Third, the research model shows four independent variables and travel intention as a dependent variable. That means all four constructs are independent and don't have any relationships. However, many previous studies show that the individual's intentional behavior and travel decision-making process is not simple causal relationships but sequential and complex relationships among variables. It should be clearly explained which theoretical background and previous studies support each relationship in the propose model.
Fourth, the result of model fit indices for the measurement model should be presented. Too many variables were deleted for the final model. That means, the measurement that was initially developed has less valid and reliable. I guess that this is caused by the lack of sound theoretical approach.
Please provided reference of model fitness judgement used in Table 5. Please provide TLI index and SRMR.
In the structural model test, why you set all correlations among independent variables? Please provide a rationale for that.
The path value of hypothesis 2 shows 0.01 not -0.01 in the figure 7. Please check it again.
The authors explain the result of hypothesis 3 test is consistent with previous studies citing 54-57. However, references of 54 and 55 is about dictionary and education study, respectively. Are you sure? How the results of the Lam & Hsu (2006) is consistent with the result of this study, in what point?
I am really doubt if the authors fully understand the theories of travel intention and this would give any theoretical contribution in tourism field.
I am sorry for the harsh expressions on your research, however I don't think this study is appropriate to be published in the highly ranked journal.
Author Response
Review 2
1.The topic of the study is interesting, however, the paper left me a number of questions throughout the reading. Overall, the paper needs to improve overall coherence and rationales of the research.
Dear reviewer: Thanks for the reminder, in the introduction, the narrative has been adjusted.
2.First, I don't see the match of the title with contents and the research purpose.
Dear reviewer: Thanks for the reminder, the title has been adjusted.
3.Second, the rationals of the research is very weak. I don't see the theoretical background and literature review is appropriate and clear in order to support the research model. For example, the authors brought tourism decision-making cognition of tourism policy decision. The study focused on travel intention of participation of the 2021 Dajia Matsu pilgrimage. Then it is better to start with theories of travel decision-making process of travelers. The authors fail to address why tourism decision-making cognition of policy is needed rather than traveler's decision making theory.
Dear reviewer: Thank you for your suggestion. In 2.1. Tourism decision-making cognition, we have strengthened the explanation of the importance of the manuscript of tourism decision-making theory.
4.Third, the research model shows four independent variables and travel intention as a dependent variable. That means all four constructs are independent and don't have any relationships. However, many previous studies show that the individual's intentional behavior and travel decision-making process is not simple causal relationships but sequential and complex relationships among variables. It should be clearly explained which theoretical background and previous studies support each relationship in the propose model.
Dear reviewer: We have made substantial corrections in the 2. Literature Discussion section, and made supplementary explanations for each theoretical background.
5.Fourth, the result of model fit indices for the measurement model should be presented. Too many variables were deleted for the final model. That means, the measurement that was initially developed has less valid and reliable. I guess that this is caused by the lack of sound theoretical approach.
Dear reviewer: We first attempt to summarize the possible topic directions based on the content of the literature review, and then use confirmatory factor analysis to obtain the convergent validity and facet validity of the questionnaire.As suggested by the literature that the MI value should be above 3.84, it should not be taken, so the issues that exceed it are deleted and the final questionnaire skeleton is obtained.
We have also strengthened the theoretical narrative in the literature, hoping to get your approval.
Related instructions such as 4.3
6.Please provided reference of model fitness judgement used in Table 5. Please provide TLI index and SRMR.
Dear reviewer: Thank you for your suggestion. We will add relevant explanations in the second paragraph of section 4.5.
7.In the structural model test, why you set all correlations among independent variables? Please provide a rationale for that.
Dear reviewer: Since there are many potential combinations within each variable and between the variable and the variable, the structural model calculation is to make assumptions on the issues first, and then verify the assumptions that are relevant to each other to obtain the most suitable issue. . Thus all the correlations between the independent variables are set.
8.The path value of hypothesis 2 shows 0.01 not -0.01 in the figure 7. Please check it again.
Dear reviewer: Thank you for your suggestions. We found a problem with the labeling in Figure 7, and we have modified it.
9.The authors explain the result of hypothesis 3 test is consistent with previous studies citing 54-57. However, references of 54 and 55 is about dictionary and education study, respectively. Are you sure? How the results of the Lam & Hsu (2006) is consistent with the result of this study, in what point?
Dear reviewer: We believe that the main reasons that attract tourists to travel will vary with personal travel motives, and the difference in importance will change with time, space, and actual conditions. In the severe epidemic situation, how to maintain physical and mental health has become the focus of tourism participation considerations. Therefore, we believe that this phenomenon is similar to the results of Lam & Hsu (2006). And in the study of Wang, Xue, Wang and Wu (2020), the influence of epidemic perception on tourism intention has been confirmed.
10.I am really doubt if the authors fully understand the theories of travel intention and this would give any theoretical contribution in tourism field.
Dear reviewer: We have adjusted the manuscript and look forward to your approval.
Finally, I sincerely thank you. We try our best to make adjustments and hope that the adjusted manuscript will be recognized. I look forward to working together to complete a sufficient level of manuscript. Thanks again for your suggestions.
Reviewer 3 Report
Very interesting paper undertaking important current problem. Nobody can predict the future development of the global pandemic Covid-19, but clearly, it impacted severely all variables connected with the tourism market. One such important variable is travel behavior and especially the travel decision-making process. Understanding factors that influence that process during pandemics is among the most important research tasks. Even, when pandemics are over the new attitudes might be present in future tourists' behavior. I appreciate strongly the research attitude in the presented paper that is focused on factors typical for Covid-19 pandemics time but can be also interpreted universally. It is a pity, that the Authors did not underline this more universal outcome of their research.
The paper is written well and there are just a few small mistakes to be corrected. Point 3.2. starts with information about 9 hypotheses, but only four are presented just below, and generally throughout the text. The research is rooted in few very wide topics of tourism analysis and they are presented in points 2.1.-2.5. In my opinion, those points are too brief. It is very difficult to find there a justification for the selection of research issues (B1-B7, C1-C8, etc.). Actually, without such a justification in chapter 2, this selection might be perceived as not justified. Additionally, it would be more convincing, if the Authors could cite more bibliography in that chapter. Most of the statements can be supported with numerous publications and usually, only one is presented.
Author Response
Review 3
1.Very interesting paper undertaking important current problem. Nobody can predict the future development of the global pandemic Covid-19, but clearly, it impacted severely all variables connected with the tourism market.
Dear reviewer: Thank you for your approval.
2.One such important variable is travel behavior and especially the travel decision-making process. Understanding factors that influence that process during pandemics is among the most important research tasks. Even, when pandemics are over the new attitudes might be present in future tourists' behavior.
Dear reviewer: Thank you for your approval.
3.I appreciate strongly the research attitude in the presented paper that is focused on factors typical for Covid-19 pandemics time but can be also interpreted universally. It is a pity, that the Authors did not underline this more universal outcome of their research.
Dear reviewer: Thanks for the reminder, we try to add in the introduction and literature review.
4.The paper is written well and there are just a few small mistakes to be corrected. Point 3.2. starts with information about 9 hypotheses, but only four are presented just below, and generally throughout the text.
Dear reviewer: Thanks for the reminder, we have fixed this error.
5.The research is rooted in few very wide topics of tourism analysis and they are presented in points 2.1.-2.5. In my opinion, those points are too brief. It is very difficult to find there a justification for the selection of research issues (B1-B7, C1-C8, etc.).
Actually, without such a justification in chapter 2, this selection might be perceived as not justified.
Dear reviewer: Thanks for the reminder, we try to add in the literature review.
6.Additionally, it would be more convincing, if the Authors could cite more bibliography in that chapter. Most of the statements can be supported with numerous publications and usually, only one is presented.
Dear reviewer: Thanks for the reminder. Regarding the content of Chapter 2, we have added information and bibliography, and reorganized it.